# Language to Rewards for Robotic Skill Synthesis

**Wenhao Yu,**\* **Nimrod Gileadi**\*, **Chuyuan Fu,**† **Sean Kirmani**†, **Kuang-Huei Lee**†,
**Montse Gonzalez Arenas, Hao-Tien Lewis Chiang, Tom Erez, Leonard Hasenclever,**
**Jan Humplik, Brian Ichter, Ted Xiao, Peng Xu, Andy Zeng, Tingnan Zhang,**
**Nicolas Heess, Dorsa Sadigh, Jie Tan, Yuval Tassa, Fei Xia**
Google DeepMind
https://language-to-reward.github.io/ ‡

**Abstract:** Large language models (LLMs) have demonstrated exciting progress in acquiring diverse new capabilities through in-context learning, ranging from logical reasoning to code-writing. Robotics researchers have also explored using LLMs to advance the capabilities of robotic control. However, since low-level robot actions are hardware-dependent and underrepresented in LLM training corpora, existing efforts in applying LLMs to robotics have largely treated LLMs as semantic planners or relied on human-engineered primitives to interface with the robot. On the other hand, reward function is a flexible representation that can be optimized for control policies to achieve diverse tasks, while their semantic richness makes them suitable to be specified by LLMs. In this work, we introduce a new paradigm that harnesses this realization by utilizing LLMs to define reward parameters that can be optimized and accomplish variety of robotic tasks. Using reward as the intermediate interface, we can effectively bridge the gap between high-level language instructions to low-level robot actions. Meanwhile, combining this with a real-time optimizer, MuJoCo MPC, empowers an interactive behavior creation experience where users can immediately observe the results and provide feedback to the system. To systematically evaluate the performance of our proposed method, we designed a total of 17 tasks for a simulated quadruped robot and a dexterous manipulator robot. We demonstrate that our proposed method reliably tackles $90\%$ of the designed tasks, while a baseline using primitive skills as the interface with Code-as-policies achieves $50\%$ of the tasks. We further validated our method on a real robot arm where complex manipulation skills such as non-prehensile pushing emerge through our interactive system.

**Keywords:** Large language model (LLM), Low-level skill learning, Legged locomotion, Dexterous manipulation

## 1 Introduction

The recent advancements in large language models (LLMs) pretrained on extensive internet data [1, 2] have revolutionized the ability to interpret and act on user inputs in natural language. These LLMs exhibit remarkable adaptability to new contexts (such as APIs [3], task descriptions [4], or textual feedback [5]), empowering tasks from logical reasoning [6, 7] to code generation [8]. These diverse applications have extended to robotics as well, where substantial progress has been made in using LLMs to drive robot behaviors [5, 4, 9, 10, 11]: from step-by-step planning [4, 9, 12], goal-oriented dialogue [10, 11], to robot-code-writing agents [3, 13]. While they impart new modes of generalization, they focus on using language to concatenate together new behaviors from an existing library of control primitives that are either manually-engineered or learned a priori. Despite having internal knowledge about robot motions, LLMs struggle with directly outputting low-level robot commands due to the limited availability of relevant training data (Fig. 1). As a result, the expression of these methods are bottlenecked by the breadth of the available primitives, the design of which often requires extensive expert knowledge or massive data collection [14, 15, 16].

To tackle these challenges, we need to operate at a level of abstraction that allows harnessing the capabilities offered by LLMs. Our key insight is to use reward functions as an interface that bridges the gap between

---

\*Co-first authors, equal contribution

†Core contributors

‡Corresponding emails: {magicmelon,nimrod,xiafei}@google.com. See Contributions in Appendix A.1

7th Conference on Robot Learning (CoRL 2023), Atlanta, USA.

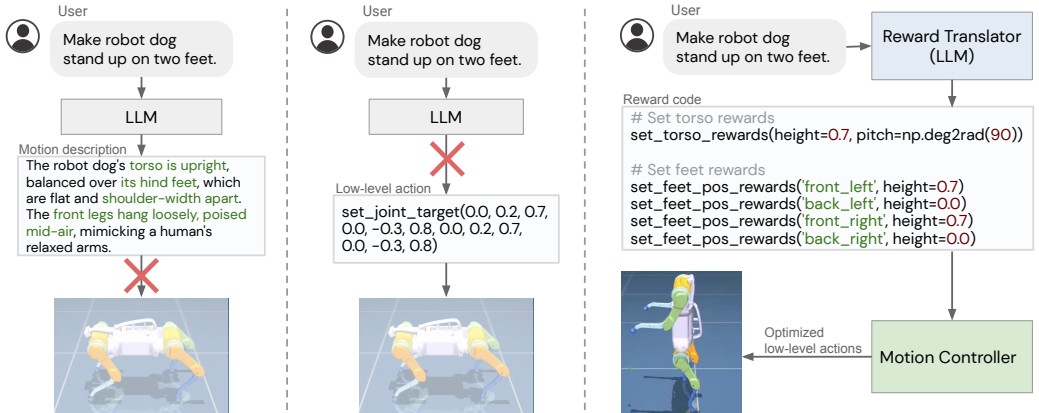

Figure 1: LLMs have some internal knowledge about robot motions, but cannot directly translate them into actions (left). Low-level action code can be executed on robots, but LLMs know little about them (mid). We bridge this gap by proposing a system (right) consisting of the Reward Translator that interprets the user input and provides a reward specification, which is then consumed by a Motion Controller that interactively synthesizes a robot motion given the reward.

language and low-level robot actions. This is motivated by the fact that language instructions from humans often tend to describe behavioral outcomes instead of low-level behavioral details (e.g. "robot standing up" versus "applying 15 Nm to hip motor"), and therefore we posit that it would be easier to connect instructions to rewards than low-level actions given the richness of semantics in rewards. In addition, reward terms are usually modular and compositional, which enables concise representations of complex behaviors, goals, and constraints. This modularity further creates an opportunity for the user to interactively steer the robot behavior. However, in many previous works in reinforcement learning (RL) or model predictive control (MPC), manual reward design requires extensive domain expertise [17, 18, 19, 20, 21]. This points to a missing link between the reward structures and task specification which is often in natural language. As such, we propose to utilize LLMs to automatically generate rewards, and leverage online optimization techniques to solve them. Concretely, we explore using LLMs to translate task semantics to reward functions, and use MuJoCo MPC, a real-time optimization tool to synthesize robot behavior in real-time [22]. Thus reward functions generated by LLMs can enable non-technical users to generate and steer novel and intricate robot behaviors without the need for vast amounts of data nor the expertise to engineer low-level primitives.

Across a span of 17 control problems on a simulated quadruped and a dexterous manipulator robot, we show that this formulation delivers diverse and challenging locomotion and manipulation skills. Examples include getting a quadruped robot to stand up, asking it to do a moonwalk, or tasking a manipulator with dexterous hand to open a faucet. We perform a large-scale evaluation to measure the overall performance of our proposed method. We compare our method to a baseline that uses a fixed set of primitive skills and an alternative formulation of grounding language to reward. We show that our proposed formulation can solve 40% more skills than baselines and is more stable in solving individual skills. We further deploy our approach to a real robot manipulator and demonstrate complex manipulation skills through language instructions.

Our work makes the following core contributions: i) We explore a novel interactive framework of using reward function as the interface to bridge language models and low-level robot actions. ii) We introduce a two-layer prompting scheme that effectively improve the reliability of the system in harnessing internal motion knowledge from LLMs. iii) We demonstrate superior performance of our proposed system to baseline methods on two robot embodiments with 17 challenging control tasks and show validation of the approach on a robotic hardware.

## 2 Related Work

**Language to Actions.** Directly predicting low-level control actions based on a language instruction has been studied using various robot learning frameworks. Early work in the language community studied mapping templated language to controllers with temporal logic [23] or learning a parser to motion primitives [24], while more recent work utilize end-to-end models that produce actions conditioned on natural language descriptions. One example is instruction following methods in navigation [25]. However, they often assume low-dimensional actions navigating from one node of the graph to another [25, 26]. To extend the end-to-end approaches to manipulation, a common approach is to utilize latent embeddings of language

commands as multitask input context, and train with behavioral cloning [14, 27, 16], offline reinforcement learning [28], goal-conditioned reinforcement learning [29], or in a shared autonomy paradigm [30]. While end-to-end trained policies can be performant, they require significant amount of data in the form of offline datasets or online environment interaction. In contrast, we study a less data hungry approach where low-level actions are not directly produced by an end-to-end policy but instead by an optimal controller.

**Language to Code.** Code generation models have been widely studied both in and outside robotics context [31, 8, 32]. The capability of those models range from solving coding competition questions [33] and benchmarks [34], to drawing simple figures [35], generating policies that solve 2D tasks [36], and complex instruction following tasks [3]. In this work, we study LLMs for generating code for reward functions, and show that the expression of the rewards can lead to expressive low-level policies.

**Language to Rewards.** The idea of translating natural language instructions to rewards has been explored by several prior work [37, 38, 39, 40, 41, 42, 43, 44]. A common strategy in this direction is to train domain-specific reward models that map language instructions to reward values [38, 42, 40] or constraints [39]. Although these methods can achieve challenging language conditioned robotic tasks such as object pushing [39], and drawer opening [40], they require considerable language-labeled data to train the reward model. Recent works investigated using LLMs directly as a reward function for inferring user intentions in negotiation games or collaborative human-AI interaction games [37, 44]. By leveraging LLMs to assign reward values during RL training, they demonstrate training agents that are aligned with user intentions and preferences without explicit reward modeling. However, these works receive reward values of rollouts when training RL policies, which requires a large number of queries to LLMs during training. In contrast, we levrage LLMs to produce a parameterized reward function that can then be optimized. A similar direction to this work is automated parameterization of reward functions, which had been explored in AutoRL [21], however, they don't provide a language interface. Finally, a concurrent work by Huang et al. explored extracting 3D value maps from large foundation models, which are used by motion planners to perform manipulation tasks [45].

**Incorporating Iterative Human Feedback.** Correcting plans with iterative language feedback has also been explored in the past. Broad et al. enable efficient online corrections using distributed correspondence graphs to ground language [46]. However, this work relies on a semantic parser with pre-defined mappings to ground language corrections. More end-to-end approaches have also demonstrated learning a language correction conditioned policy, but they are similarly data hungry and thus fall back to shared autonomy to reduce complexity [47]. Later work explore mapping language corrections to composable cost functions similar to our work by training a prediction model from demonstration and apply trajectory optimization to perform control [39]. Followup works further simplifies the system by integrating language corrections to directly modify the waypoints of a trajectory using extensive datasets of paired corrections and demonstrations [48, 49]. In contrast to these prior work, we demonstrate a flexible and data-efficient approach that leverages LLMs to allow for multi-step correction of reward functions based on human feedback.

## 3 Grounding Language to Actions Using Rewards

### 3.1 Background and Reward Interface

Our system takes user instruction in natural language and synthesizes corresponding reward functions for the desired motion. We define the reward function in the context of Markov Decision Process (MDP): $(S, A, R, P, p_0)$, where $S$ is the state space, $A$ is the action space, $R : S \times A \mapsto \mathbb{R}$ is the reward function, $P : S \times A \mapsto S$ is the dynamics, and $p_0$ is the initial state distribution. Given a reward function $R$, an optimal controller finds a sequence of actions $\mathbf{a}_{1:H} = \{\mathbf{a}_1, ..., \mathbf{a}_H\}$ that maximizes the expected accumulated reward: $J(\mathbf{a}_{1:H}) = \mathbb{E}_{\tau=(\mathbf{s}_0, \mathbf{a}_0, ..., \mathbf{s}_H)} \sum_{t=0}^{H} R(\mathbf{s}_t, \mathbf{a}_t)$, where $H$ is the rollout horizon.

In this work, we assume the reward takes a particular form, suitable for use with MJPC. The reward is the sum of a set of individual terms:

$$R(\mathbf{s}, \mathbf{a}) = -\sum_{i=0}^{M} w_i \cdot \mathrm{n}_i \big( r_i(\mathbf{s}, \mathbf{a}, \psi_i) \big), \tag{1}$$

where $w \in \mathbf{R}_+$ is a non-negative weight, $\mathrm{n}(\cdot) : \mathbf{R} \to \mathbf{R}_+$ is a twice-differentiable norm that takes its minimum at 0, $r \in \mathbf{R}$ is a residual term that achieves optimality when $r = 0$, and $\psi_i$ is the parameters of the $i_{th}$ residual term. For example, if we want to have the robot raise its body height $h$ to a desired height, we may design a residual term $r_h(h, \psi) = h - \psi$, where the reward parameter $\psi$ denotes the desired height, and use the l2 norm to construct the final reward function: $R_h = -w \|r_h\|_2$. In principle, one may design

task-specific residual terms that can solve particular controller tasks. However, designing these residuals requires domain expertise and may not generalize to novel tasks. In this work, we use a set of generic and simple residual terms, and leverage the power of LLMs to compose different terms to generate complex behaviors. The full set of residual terms used in this work can be found in the Appendix A.6.

Our proposed system consists of two key components (Fig. 1 right): i) a *Reward Translator*, built upon pre-trained Large Language Models (LLMs) [10], that interacts with and understands user intents and modulates all reward parameters $\psi$ and weights $w$, and ii) a *Motion Controller*, based on MuJoCo MPC [22], that takes the generated reward and interactively optimize the optimal action sequence $\mathbf{a}_{1:H}$. Below we provide more details on the design of Reward Translator and Motion Controller.

## 3.2 Reward Translator

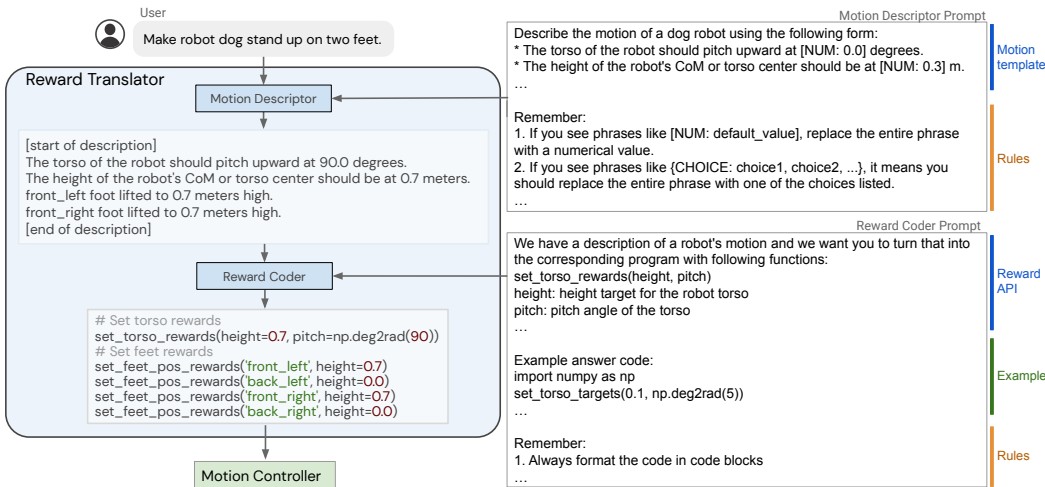

Figure 2: Detailed dataflow of the Reward Translator. A *Motion Descriptor* LLM takes the user input and describe the user-specified motion in natural language, and a *Reward Coder* translates the motion into the reward parameters.

Inspired by recent progress on Large Language Models (LLMs), we propose to build the Reward Translator based on LLMs to map user interactions to reward functions corresponding to the desired robot motion. As reward tuning is highly domain-specific and requires expert knowledge, it is unsurprising that LLMs trained on generic language datasets (e.g. [1]) cannot directly generate a reward for a specific hardware. Instead, we explore the in-context learning ability of LLMs to achieve this goal, inspired by prior work that demonstrated a variety of in-context learning skills for LLMs [2, 50]. Furthermore, we decompose the problem of language to reward into two stages: motion description and reward coding task, as illustrated in Fig. 2.

**Motion Description** In the first stage, we design a *Motion Descriptor* LLM that interprets and expands the user input into a natural language description of the desired robot motion following a pre-defined template (see example in Fig. 2). Although it is possible for LLMs to directly generate reasonable reward functions for relatively simple task, it often fails for tasks that necessitates complex reasoning. On the other hand, as observed in Fig. 1 left, LLMs can describe complex motions in detailed natural language successfully.

Inspired by this observation, we design a template that describes common movements of a robot (see Fig. 2 top right for an example) to effectively harness LLMs' internal knowledge about motions. *Motion Descriptor* produces more structured and predictable outputs and improves stability of the overall system. In addition, as we are describing the motion in natural language, we do not need to provide any specific examples in the prompt and can rely entirely on LLMs to generate the result.

**Reward Coding** In the second stage, we translate the generated motion description into the reward function using a second LLM. We formulate the problem of language to reward function as a code-writing task to benefit from the LLMs' knowledge of coding and code structure, thus we name the second LLM the *Reward Coder*. We design a prompt for instructing the LLM to generate reward specifying code (see example in Fig. 2 bottom right). The prompt consists of three parts: i) description of the reward APIs that the LLM can call to specify different parameters of the reward function, ii) an example response that we expect the *Reward Coder* to produce, and iii) the constraints and rules that the *Reward Coder* needs

to follow. Note that the example is to demonstrate to the LLM how the response should look like, instead of teaching it how to perform a specific task. As such, the *Reward Coder* needs to specify the reward parameters based on its own knowledge about the motion from the natural language description.

## 3.3 Motion Controller

The Motion Controller needs to map the reward function generated by the Reward Translator to low-level robot actions $\mathbf{a}_{1:H}$ that maximize the accumulated reward $J(\mathbf{a}_{1:H})$. There are a few possible ways to achieve this, including using reinforcement learning (RL), offline trajectory optimization, or, as in this work, model predictive control (MPC). At each control step, MPC plans a sequence of optimized actions $\mathbf{a}_{1:H}$ and sends to the robot. The robot applies the action corresponding to its current timestamp, advances to the next step, and sends the updated robot states to the MPC planner to initiate the next planning cycle. The frequent re-planning in MPC empowers its robustness to uncertainties in the system and, importantly, enables interactive motion synthesis and correction. We use an open-source implementation based on the MuJoCo simulator [51], MJPC [22]. MJPC has demonstrated the interactive creation of diverse behaviors such as legged locomotion, grasping, and finger-gaiting while supporting multiple planning algorithms, such as iLQG and Predictive Sampling. Following the observation by Howell et al [22], second-order planners such as iLQG produces smoother and more accurate actions while zeroth-order planners such as Predictive Sampling is better at exploring non-smooth optimization landscape, we use iLQG for legged locomotion tasks, while use Predictive Sampling for manipulation tasks in this work.

# 4 Experiments

We design experiments to answer the following questions: 1) Is our proposed method, by combining LLMs and MJPC, able to generate diverse and complex robot motions through natural language interface? 2) Does interfacing with the reward function result in a more expressive pipeline than interfacing directly with low-level or primitive actions and is Motion Descriptor necessary for achieving reliable performance? 3) Can our method be applied to real robot hardware?

## 4.1 Experiment Setup

We first evaluate our approach on two simulated robotic systems: a quadruped robot, and a dexterous robot manipulator (Fig. 3). Both robots are modeled in MuJoCo MPC [22]. We use GPT-4 as the underlying LLM module with temperature parameter set to 0.3 [52]. Here we describe the key setups of each robot. More details regarding the full prompts and reward function can be found in Appendix A.5 and A.6.

**Quadruped Robot** In this example, we command a four legged, 12 DoF robot (Fig. 3 (a)) to perform a variety of motor skills. Quadruped robots have been demonstrated to perform a large variety of skills including locomotion [35], hopping [18], biped standing [53, 54], parkour [55] etc. We apply our system to the quadruped robot to perform a similar suite of skills while only using natural language as input.

**Dexterous Manipulator** In the second example, we demonstrate our system on a dexterous manipulator robot consisting of a 7 DoF Franka Emika arm and a 20 DoF shadow hand as the end-effector (Fig. 3 (b)). This creates a large action space, making it challenging to manually design a controller for this robot.

## 4.2 Baselines

We compare our proposed system to two baseline methods: i) an variant of our approach that only uses *Reward Coder* without having access to the *Motion Descriptor*, and ii) Code-as-Policies [3] where the LLM generates a plan using a set of pre-defined primitive skills. For the Code-as-Policies (CaP) baseline, we design the primitive skills based on common commands available to the robot. Due to limited space we put the full list of primitives in Appendix A.3.

## 4.3 Tasks

We design nine tasks for the quadruped robot and eight tasks for the dexterous manipulator to evaluate the performance of our system. Fig. 3 shows samples of the tasks. The full list of tasks and sampled videos can be found in Appendix A.2 and project website [4].

---

[4] `language-to-reward.github.io`

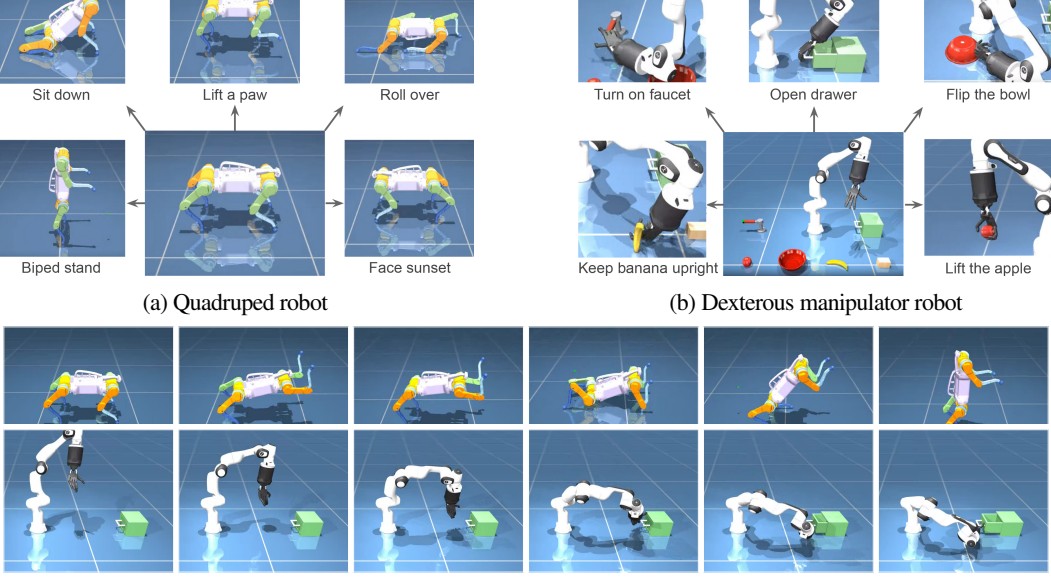

(a) Quadruped robot          (b) Dexterous manipulator robot

(c) Example rollout for the two robots.

Figure 3: The two robots used in our experiments and sampled tasks. (a) a Quadruped robot with 12 DoFs. (b) a dexterous manipulator robot with 27 DoFs. (c) example rollouts produced by our algorithm.

## 4.4 Evaluation results

For each task and method considered, we generate 10 responses from Reward Translator , each evaluated in MJPC for 50 times. Fig. 4 shows the results for both robots. Our proposed approach achieves notably higher success rate for $11/17$ task categories and comparable performance for the rest tasks, showing the effectiveness of the proposed method. When compared to the CaP baseline, our method achieves better success rate in almost all tasks. This is due to that CaP can perform well on tasks that can be expressed by the given primitives (e.g. Touch object) or very close to the given examples in prompt (e.g. Sit down), but fails to generalize to novel skills. On the other hand, using Reward Coder only can achieve success on some tasks but fails in ones that requires more reasoning. For example, when asked to open a drawer, the baseline often forget to task the robot hand to get closer to the drawer handle and only design the reward for encouraging the drawer to be open. Sampled responses from different method can be found in Appendix A.8.

To further understand the overall performance of different systems, we show the pass rate in Fig. 4 right, which is a standard metric for analyzing code generation performance [8]. Each point in the plot represents the percentage of tasks the system can solve, given that it can generate N pieces of code for each task and pick the best performing one. As such, the pass rate curve measures the stability of the system (the more flat it is, the more stable the system is) as well as the task coverage of the system (the converged point represents how many tasks the system can solve given sufficient trials). It is clear from the result that for both embodiments, using reward as the interface empowers LLMs to solve more tasks more reliably, and the use of Structured Motion Description further boosts the system performance.

## 4.5 Ablation Study

In this section, we perform an ablation study to investigate how our proposed pipeline responds to varying amount of rules and reminders provided to Reward Translator and Motion Controller . Specifically, for Reward Translator , we removed all rules in the prompt except for rule 5 that specifies the output format. Similarly for Motion Controller , we removed all rules except for rule 1 to make sure LLMs respond in the correct format. We compare our full prompt to three variants: 1) reduced rule for Reward Translator only, 2) reduced rule for Motion Controller , 3) reduced rule for both modules. The results are shown in Figure 5. We can see that the full prompts outperforms the ablated variants, showing the usefulness of the rules. Furthermore, we find reducing the rules in the Motion Controller caused a larger performance degradation than reducing the rules in Reward Translator .

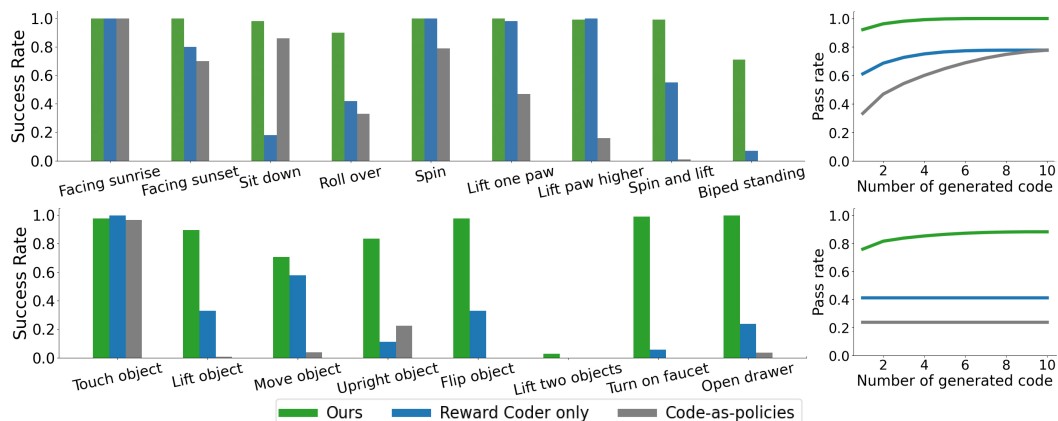

Figure 4: Comparison of our method and alternative methods in terms of pass rate: if we generate N pieces of code for each task and pick the best performing one, what's the percentage of tasks that the system can successfully tackle.

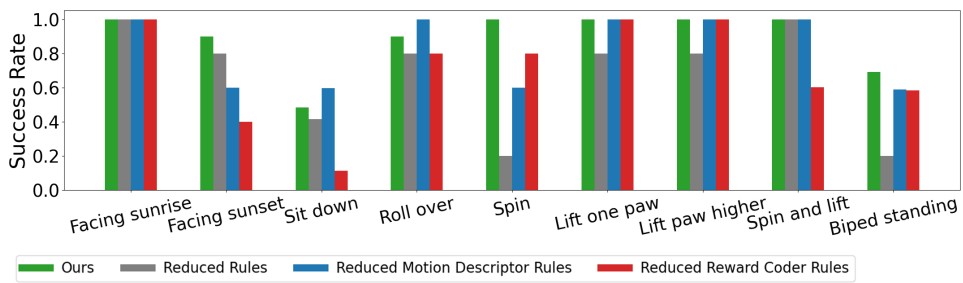

Figure 5: Ablation study for different amount of rules in the designed prompts.

## 4.6 Interactive Motion Synthesis Results

One benefit of using a real time optimization tool like MJPC is that humans can observe the motion being synthesized in real time and provide feedback. We showcase two examples where we teach the robot to perform complex tasks through multiple rounds of interactions. In the first example, we task the quadruped robot to stand up and perform a moon-walk skill (Fig. 6a). We give four instructions to achieve the task, as shown in Fig. 6. Each instruction improves the behavior towards the desired behavior based on the interactively synthesized results. This showcase that users can interactively shape the behavior of the robot in natural language. In the second example, we showcase a different way of leveraging the interactivity of our system by sequentially commanding the dexterous manipulator robot to place an apple in a drawer, as seen in Fig. 6b. Results of the interactive results are best viewed in the supplementary video and full code output from our method can be found in Appendix A.9.

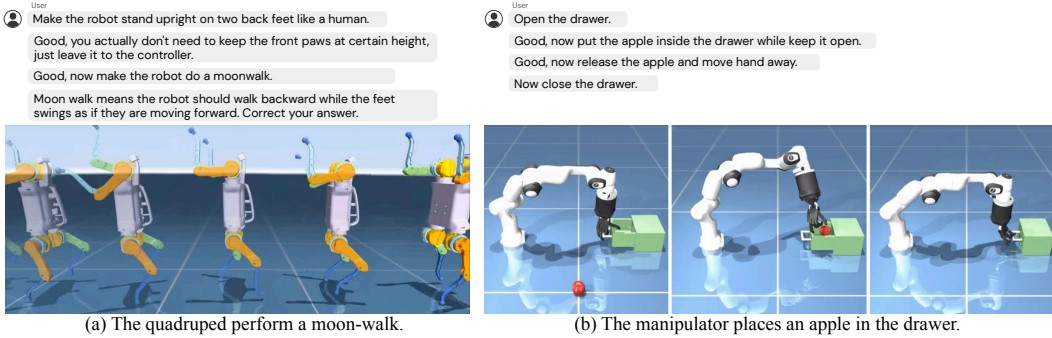

(a) The quadruped perform a moon-walk.    (b) The manipulator places an apple in the drawer.

Figure 6: The two interactive examples using our proposed system.

## 4.7 Real-robot experiments

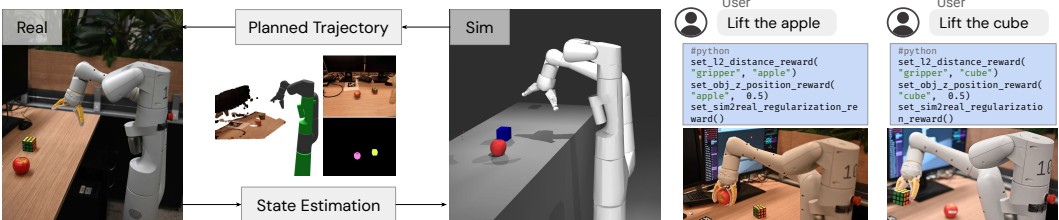

Figure 7: Implementation and rollouts of the proposed system in the real world.

We implement a version of our method on a mobile manipulator, and tested it on nonprehensile manipulation tasks in the real world. To obtain object states in real-world, we first use an open-vocabulary detector: F-VLM [56] to segment the object and then extract the associated points from point cloud behind the mask and perform outlier rejection for points that might belong to the background. From a birds-eye view, we fit a minimum volume rectangle and take the extremes to determine the extent in the z-axis. We use this 3D bounding box as state estimation for corresponding object in simulation. To detect the surface of the table with proper orientation, we use an AprilTag [57]. In addition, as seen in the supplementary video, MJPC sometimes discover highly dexterous and dyanmic maneuvers to accomplish the desired task that are beyond the capabilities of current real hardwares. To mitigate this issue, we design a regularization residual term specific to encourage steady and stable robot movements when applying our system to the real robot (set_sim2real_regularization_reward() in Fig. 7, see Appendix A.6.3 for details for this term).

We demonstrate sim-to-real transfer on three tasks: object pushing, grasping, and drawer opening. Our system is able to generate relevant reward code and synthesize the correct motion. We also measured success rate of our system on three scenarios: 1) picking up an apple, 2) picking up a rubic's cube, and 3) opening the middle drawer by repeating our approach for 10 time. Our method achieved 70%, 70%, 80% success rate respectively. For example rollouts please refer to the supplementary video/website and Fig. 8.

## 5 Discussion and Conlusion

We propose a new paradigm for interfacing an LLM with a robot through reward functions, powered by a low-level model predictive control tool, MuJoCo MPC. Using reward function as the interface enables LLMs to work in a semantic-rich space that play to the strengths of LLMs, while ensures the expressiveness of the resulting controller. To further improve the performance of the system, we propose to use a motion description template to better extract internal knowledge about robot motions from LLMs. We evaluate our proposed system on two simulated robotic platforms: a quadruped robot and a dexterous manipulator robot. We apply our approach to both robots to acquire a wide variety of skills. Compared to alternative methods that do not use reward as the interface, or do not use the motion description template, our method achieves significantly better performance in terms of stability and the number of tasks it can solve.

**Limitations and Future Work.** Though we show that our system can obtain a diverse set of skills through natural language interactions, there are a few limitations. First, we currently design templates of motion descriptions for each type of robot morphology, which requires manual work. An interesting future direction is to unify or automate the template design to make the system easily extendable to novel robot morphologies. Second, our method currently relies on language as the interaction interface with human users. As such, it can be challenging to design tasks that are not easily described in language (e.g., "walk gracefully"). One potential way to mitigate this issue is to extend the system to multi-modal inputs to allow richer forms of user interactions (e.g., by showing a video of the desirable behavior). Thirdly, we currently use pre-defined reward terms whose weights and parameters are modulated by the LLMs. Constraining the reward design space helps improve stability of the system while sacrifices some flexibility. For example, our current design does not support time-varying rewards and would require re-designing the prompt to support that. Enabling LLMs to reliably design reward functions from scratch is thus an important and fruitful research direction.

### Acknowledgments

The authors would like to acknowledge Ken Caluwaerts, Kristian Hartikainen, Steven Bohez, Carolina Parada, Marc Toussaint, and the greater teams at Google DeepMind for their feedback and contributions.

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

# A  Appendix

## A.1  Author Contributions

Author contributions by type, ordered alphabetically within each category:

**Method (conception, implementation, iteration, evaluation)**: Nimrod Gileadi, Kuang-Huei Lee, Yuval Tassa, Fei Xia, Peng Xu, Wenhao Yu.

**Infrastructure Development**: Tom Erez, Nimrod Gileadi, Yuval Tassa, Fei Xia, Wenhao Yu.

**Hardware Deployment**: Chuyuan Fu, Nimrod Gileadi, Leonard Hasenclever, Jan Humplik, Sean Kirmani, Yuval Tassa, Fei Xia, Ted Xiao, Wenhao Yu.

**Project Advising**: Tom Erez, Nicolas Heess, Brian Ichter, Dorsa Sadigh, Jie Tan, Yuval Tassa, Fei Xia, Andy Zeng, Tingnan Zhang.

**Paper Writing/Revision**: Montse Gonzalez Arenas, Hao-Tien Lewis Chiang, Nimrod Gileadi, Brian Ichter, Dorsa Sadigh, Fei Xia, Andy Zeng, Wenhao Yu, Tingnan Zhang.

## A.2  Full task list

Here we show the list of tasks used in our evaluation as well as the instructions used for each task.

| Task | Instructions | Expected Behavior |
|---|---|---|
| Facing sunrise | It's early in the morning, make the robot head towards the sun. | Robot face towards East. |
| Facing sunset | It's late in the afternoon, make the robot head towards the sunset. | Robot face towards West. |
| Sit down | Sit down low to ground with torso flat. | Robot's CoM drops lower and remain flat. |
| Roll Over | I want the robot to roll by 180 degrees. | Robot's belly faces up. |
| Spin | Spin fast. | Robot reach a fast turning speed. |
| Lift one paw | I want the robot to lift its front right paw in the air. | The front right paw of the robot lifts up in the air. |
| Lift paw higher | I want the robot to lift its front right paw in the air. Lift it even higher. | The robot lifts its front right paw higher than before. |
| Spin with lifted paws | Lift front left paw. Good, now lift diagonal paw as well. Good, in addition I want the robot to spin fast. | Robot lifts front left and rear right paws while spin fast. |
| Stand up on two feet | Make the robot stand upright on two back feet like a human. | Robot stands on two back feet and keep balance. |
| Trotting | Describe trotting gait, then make the robot walk in a trotting gait. | Robot walks with diagonal feet in synchronous. |
| Pacing | Describe pacing gait, then make the robot walk in a pacing gait. | Robot walks with feet on the same side in synchronous. |
| Bounding | Describe bounding gait, then make the robot walk in a bounding gait. | Robot walks with front and back feet in synchronous respectively. |

Table 1: List of tasks used in evaluation for the quadruped robot.

| Task | Instructions | Expected Behavior |
|---|---|---|
| Touch object | Touch the {object} | Robot fingers in contact with the object. |
| Lift object | Lift the {object} to 0.5 m | The object needs to stay above 0.4 m for 1s. |
| Move object | Move the {object_a} to {object_b} | The distance between object needs to be smaller than 0.1 m. |
| Upright object | Place the {object} upright | The z axis of the object needs to be parallel to x-y plane. |
| Flip object | Flip the {object} | The local up vector of the object should be pointing downward. |
| Lift two objects | Lift the {object_a} and {object_b} at the same time. | Both objects need to stay above 0.4 m for 1 s. |
| Turn on the faucet | Turn on the faucet. | The valve of the faucet needs to be turned 90 degrees. |
| Open the drawer | Open the drawer. | The drawer needs to be pulled fully open. |

Table 2: List of tasks used in evaluation for the dexterous manipulation.

## A.3  Baseline details

For the quadruped robot, we use the following three primitive skills:

- `head_towards(direction)` specifies a target heading direction `direction` for the robot to reach.
- `walk(forward_speed, sideway_speed, turning_speed)` controls the robot to walk and turn in different directions. This is a common interface used in quadruped robots to navigate in different environments.
- `set_joint_poses(leg_name, joint_angles)` directly sets the joint positions for each DoF on the robot. To help the LLMs understand the joint angles, we provide a set of examples in the prompt.

For the dexterous manipulator robot, we use three primitive skills to control the robot motion and also a function to get access to the position of an object in the scene:

- `end_effector_to(position)` moves the center of the robot hand's palm to the given `position`.
- `end_effector_open()` opens the hand of the robot by extending all fingers.
- `end_effector_close()` closes the hand to form a grasping pose.
- `get_object_position(obj_name)` gets the position of a certain object in the scene.
- `get_joint_position(joint_name)` gets the position of a certain joint in the scene.

## A.4  Additional illustrations for real-world results

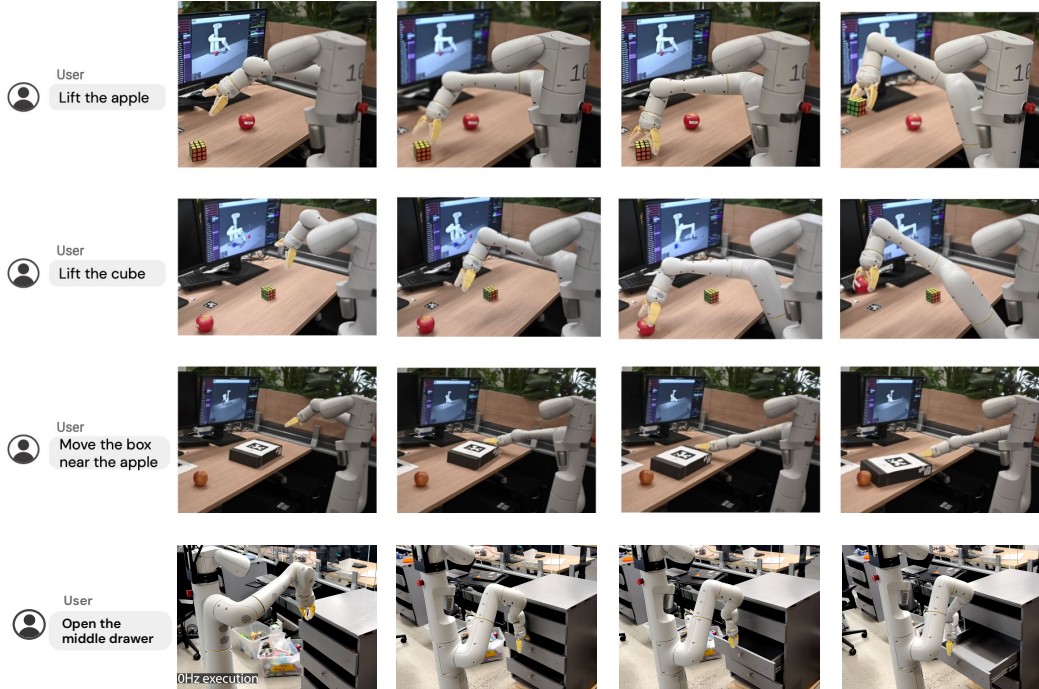

Figure 8: More illustrations for the real-world results for the proposed system.

## A.5  Full Prompts

Here we list the full prompts used in *Reward Translator* for all experiments used in this work.

**i) Motion Descriptor Prompt for Quadruped**

Describe the motion of a dog robot using the following form:
[start of description]
The torso of the robot should roll by [NUM: 0.0] degrees towards right, the torso should pitch upward at [NUM: 0.0] degrees.
The height of the robot's CoM or torso center should be at [NUM: 0.3] meters.
The robot should {CHOICE: [face certain direction, turn at certain speed]}. If facing certain direction, it should be facing {CHOICE: [east, south, north, west]}. If turning, it should turn at [NUM: 0.0] degrees/s.
The robot should {CHOICE: [go to a certain location, move at certain speed]}. If going to certain location, it should go to (x=[NUM: 0.0], y=[NUM: 0.0]). If moving at certain speed, it should move forward at [NUM: 0.0]m/s and sideways at [NUM: 0.0]m/s (positive means left).
[optional] front_left foot lifted to [NUM: 0.0] meters high.
[optional] back_left foot lifted to [NUM: 0.0] meters high.
[optional] front_right foot lifted to [NUM: 0.0] meters high.
[optional] back_right foot lifted to [NUM: 0.0] meters high.
[optional] front_left foot extend forward by [NUM: 0.0] meters.
[optional] back_left foot extend forward by [NUM: 0.0] meters.
[optional] front_right foot extend forward by [NUM: 0.0] meters.
[optional] back_right foot extend forward by [NUM: 0.0] meters.
[optional] front_left foot shifts inward laterally by [NUM: 0.0] meters.
[optional] back_left foot shifts inward laterally by [NUM: 0.0] meters.
[optional] front_right foot shifts inward laterally by [NUM: 0.0] meters.
[optional] back_right foot shifts inward laterally by [NUM: 0.0] meters.
[optional] front_left foot steps on the ground at a frequency of [NUM: 0.0] Hz, during the stepping motion, the foot will move [NUM: 0.0] meters up and down, and [NUM: 0.0] meters forward and back, drawing a circle as if it's walking {CHOICE: forward, back}, spending [NUM: 0.0] portion of the time in the air vs gait cycle.
[optional] back_left foot steps on the ground at a frequency of [NUM: 0.0] Hz, during the stepping motion, the foot will move [NUM: 0.0] meters up and down, and [NUM: 0.0] meters forward and back, drawing a circle as if it's walking {CHOICE: forward, back}, spending [NUM: 0.0] portion of the time in the air vs gait cycle.
[optional] front_right foot steps on the ground at a frequency of [NUM: 0.0] Hz, during the stepping motion, the foot will move [NUM: 0.0] meters up and down, and [NUM: 0.0] meters forward and back, drawing a circle as if it's walking {CHOICE: forward, back}, spending [NUM: 0.0] portion of the time in the air vs gait cycle.
[optional] back_right foot steps on the ground at a frequency of [NUM: 0.0] Hz, during the stepping motion, the foot will move [NUM: 0.0] meters up and down, and [NUM: 0.0] meters forward and back, drawing a circle as if it's walking {CHOICE: forward, back}, spending [NUM: 0.0] portion of the time in the air vs gait cycle.
[optional] The phase offsets for the four legs should be front_left: [NUM: 0.0], back_left: [NUM: 0.0], front_right: [NUM: 0.0], back_right: [NUM: 0.0].
[end of description]
Rules:
1. If you see phrases like [NUM: default_value], replace the entire phrase with a numerical value.
2. If you see phrases like CHOICE: [choice1, choice2, ...], it means you should replace the entire phrase with one of the choices listed. Be sure to replace all of them. If you are not sure about the value, just use your best judgement.
3. Phase offset is between [0, 1]. So if two legs' phase offset differs by 0 or 1 they are moving in synchronous. If they have phase offset difference of 0.5, they are moving opposite in the gait cycle.
4. The portion of air vs the gait cycle is between [0, 1]. So if it's 0, it means the foot will always stay on the ground, and if it's 1 it means the foot will always be in the air.
5. I will tell you a behavior/skill/task that I want the quadruped to perform and you will provide the full description of the quadruped motion, even if you may only need to change a few lines. Always start the description with [start of description] and end it with [end of description].
6. We can assume that the robot has a good low-level controller that maintains balance and stability as long as it's in a reasonable pose.
7. You can assume that the robot is capable of doing anything, even for the most challenging task.
8. The robot is about 0.3m high in CoM or torso center when it's standing on all four feet with horizontal body. It's about 0.65m high when it stand upright on two feet with vertical body. When the robot's torso/body is flat and parallel to the ground, the pitch and roll angles are both 0.
9. Holding a foot 0.0m in the air is the same as saying it should maintain contact with the ground.
10. Do not add additional descriptions not shown above. Only use the bullet points given in the template.

11. If a bullet point is marked [optional], do NOT add it unless it's absolutely needed.
12. Use as few bullet points as possible. Be concise.

## ii) Reward Coder Prompt for Quadruped

We have a description of a robot's motion and we want you to turn that into the corresponding program with following functions:

```
def set_torso_targets(target_torso_height,
    target_torso_pitch, target_torso_roll, target_torso_location_xy,
    target_torso_velocity_xy, target_torso_heading, target_turning_speed)
```

target_torso_height: how high the torso wants to reach. When the robot is standing on all four feet in a normal standing pose, the torso is about 0.3m high.
target_torso_pitch: How much the torso should tilt up from a horizontal pose in radians. A positive number means robot is looking up, e.g. if the angle is 0.5*pi the robot will be looking upward, if the angel is 0, then robot will be looking forward.
target_torso_velocity_xy: target torso moving velocity in local space, x is forward velocity, y is sideway velocity (positive means left).
target_torso_heading: the desired direction that the robot should face towards. The value of target_torso_heading is in the range of 0 to 2*pi, where 0 and 2*pi both mean East, pi being West, etc.
target_turning_speed: the desired turning speed of the torso in radians per second.
Remember: one of target_torso_location_xy and target_torso_velocity_xy must be None. one of target_torso_heading and target_turning_speed must be None. No other inputs can be None.

```
def set_feet_pos_parameters(feet_name,
    lift_height, extend_forward, move_inward)
```

feet_name is one of ("front_left", "back_left", "front_right", "back_right").
lift_height: how high should the foot be lifted in the air. If is None, disable this term. If it's set to 0, the foot will touch the ground.
extend_forward: how much should the foot extend forward. If is None, disable this term.
move_inward: how much should the foot move inward. If is None, disable this term.

```
def set_feet_stepping_parameters(feet_name, stepping_frequency, air_ratio,
    phase_offset, swing_up_down, swing_forward_back, should_activate)
```

feet_name is one of ("front_left", "rear_left", "front_right", "rear_right").
air_ratio (value from 0 to 1) describes how much time the foot spends in the air versus the whole gait cycle. If it's 0 the foot will always stay on ground, and if it's 1 it'll always stay in the air.
phase_offset (value from 0 to 1) describes how the timing of the stepping motion differs between different feet. For example, if the phase_offset between two legs differs by 0.5, it means one leg will start the stepping motion in the middle of the stepping motion cycle of the other leg. swing_up_down is how much the foot swings vertical during the motion cycle.
swing_forward_back is how much the foot swings horizontally during the motion cycle. If swing_forward_back is positive, the foot would look like it's going forward, if it's negative, the foot will look like it's going backward.
If should_activate is False, the leg will not follow the stepping motion.

```
def execute_plan(plan_duration=2)
```

This function sends the parameters to the robot and execute the plan for "plan_duration" seconds, default to be 2
Example answer code:

```python
import numpy as np  # import numpy because we are using it below

reset_reward()
    # This is a new task so reset reward; otherwise we don't need it
set_torso_targets(0.1,
    np.deg2rad(5), np.deg2rad(15), (2, 3), None, None, np.deg2rad(10))

set_feet_pos_parameters("front_left", 0.1, 0.1, None)
set_feet_pos_parameters("back_left", None, None, 0.15)
set_feet_pos_parameters("front_right", None, None, None)
set_feet_pos_parameters("back_right", 0.0, 0.0, None)
```

```
    set_feet_stepping_parameters("front_right", 2.0, 0.5, 0.2, 0.1, -0.05, True)
    set_feet_stepping_parameters("back_left", 3.0, 0.7, 0.1, 0.1, 0.05, True)
    set_feet_stepping_parameters("front_left", 0.0, 0.0, 0.0, 0.0, 0.0, False)
    set_feet_stepping_parameters("back_right", 0.0, 0.0, 0.0, 0.0, 0.0, False)

    execute_plan(4)
```

Remember: 1. Always format the code in code blocks.
2. Do not invent new functions or classes. The only allowed functions you can call are the ones listed above. Do not leave unimplemented code blocks in your response.
3. The only allowed library is numpy. Do not import or use any other library. If you use np, be sure to import numpy.
4. If you are not sure what value to use, just use your best judge. Do not use None for anything.
5. Do not calculate the position or direction of any object (except for the ones provided above). Just use a number directly based on your best guess.
6. For set_torso_targets, only the last four arguments (target_torso_location_xy, target_torso_velocity_xy, target_torso_heading, target_turning_speed) can be None. Do not set None for any other arguments.
7. Don't forget to call execute_plan at the end.

### iii) Baseline: Code-as-Policies Prompt for Quadruped

We have a quadruped robot. It has 12 joints in total, three for each leg. We can use the following functions to control its movements:

```
def set_target_joint_angles(leg_name, target_joint_angles)
```

leg_name is one of ("front_left", "back_left", "front_right", "back_right").
target_joint_angles: a 3D vector that describes the target angle for the abduction/adduction, hip, and knee joint of the each leg.

```
def walk(forward_speed, sideway_speed, turning_speed)
```

forward_speed: how fast the robot should walk forward
sideway_speed: how fast the robot should walk sideways
turning_speed: how fast the robot should be turning (positive means turning right)

```
def head_towards(heading_direction)
```

heading_direction: target heading for the robot to reach, in the range of 0 to 2pi, where 0 means East, 0.5pi means North, pi means West, and 1.5pi means South.

```
def execute_plan(plan_duration=10)
```

This function sends the parameters to the robot and execute the plan for "plan_duration" seconds, default to be 2
Details about joint angles of each leg: abduction/adduction joint controls the upper leg to swinging inward/outward. When it's positive, legs will swing outward (swing to the right for right legs and left for left legs). When it's negative, legs will swing inward.
hip joint controls the upper leg to rotate around the shoulder. When it's zero, the upper leg is parallel to the torso (hip is same height as shoulder), pointing backward. When it's positive, the upper leg rotates downward so the knee is below the shoulder. When it's 0.5pi, it's perpendicular to the torso, pointing downward. When it's negative, the upper leg rotates upward so the knee is higher than the shoulder.
knee joint controls the lower leg to rotate around the knee. When it's zero, the lower leg is folded closer to the upper leg. knee joint angle can only be positive. When it's 0.5pi, the lower leg is perpendicular to the upper leg. When it's pi, the lower leg is fully streching out and parallel to the upper leg.
Here are a few examples for setting the joint angles to make the robot reach a few key poses: standing on all four feet:

```
set_target_joint_angles("front_left", [0, 1, 1.5])
set_target_joint_angles("back_left", [0, 0.75, 1.5])
set_target_joint_angles("front_right", [0, 1, 1.5])
set_target_joint_angles("back_right", [0, 0.75, 1.5])
execute_plan()
```

sit down on the floor:

```
set_target_joint_angles("front_left", [0, 0, 0])
```

```
    set_target_joint_angles("back_left", [0, 0, 0])
    set_target_joint_angles("front_right", [0, 0, 0])
    set_target_joint_angles("back_right", [0, 0, 0])
    execute_plan()
```

lift front left foot:

```
    set_target_joint_angles("front_left", [0, 0.45, 0.35])
    set_target_joint_angles("back_left", [0, 1, 1.5])
    set_target_joint_angles("front_right", [0, 1.4, 1.5])
    set_target_joint_angles("back_right", [0, 1, 1.5])
    execute_plan()
```

lift back left foot:

```
    set_target_joint_angles("front_left", [0, 0.5, 1.5])
    set_target_joint_angles("back_left", [0, 0.45, 0.35])
    set_target_joint_angles("front_right", [0, 0.5, 1.5])
    set_target_joint_angles("back_right", [0, 0.5, 1.5])
    execute_plan()
```

Remember:
1. Always start your response with [start analysis]. Provide your analysis of the problem within 100 words, then end it with [end analysis].
2. After analysis, start your code response, format the code in code blocks.
3. Do not invent new functions or classes. The only allowed functions you can call are the ones listed above. Do not leave unimplemented code blocks in your response.
4. The only allowed library is numpy. Do not import or use any other library. If you use np, be sure to import numpy.
5. If you are not sure what value to use, just use your best judge. Do not use None for anything.
6. Do not calculate the position or direction of any object (except for the ones provided above). Just use a number directly based on your best guess.
7. Write the code as concisely as possible and try not to define additional variables.
8. If you define a new function for the skill, be sure to call it somewhere.
9. Be sure to call execute_plan at the end.

## iv) Motion Descriptor Prompt for Dexterous Manipulator

We have a dexterous manipulator and we want you to help plan how it should move to perform tasks using the following template:
[start of description]
To perform this task, the manipulator's palm should move close to {CHOICE: apple, banana, box, bowl, drawer_handle, faucet_handle, drawer_center, rest_position}.
object1={CHOICE: apple, banana, box, bowl, drawer_handle, faucet_handle, drawer_center} should be close to object2={CHOICE: apple, banana, box, bowl, drawer_handle, faucet_handle, drawer_center, nothing}.
[optional] object1 needs to be rotated by [NUM: 0.0] degrees along x axis.
[optional] object2 needs to be rotated by [NUM: 0.0] degrees along x axis.
[optional] object1 needs to be lifted to a height of [NUM: 0.0]m at the end.
[optional] object2 needs to be lifted to a height of [NUM: 0.0]m at the end.
[optional] object3={CHOICE: drawer, faucet} needs to be {CHOICE: open, closed}.
[end of description]
Rules:
1. If you see phrases like [NUM: default_value], replace the entire phrase with a numerical value.
2. If you see phrases like {CHOICE: choice1, choice2, ...}, it means you should replace the entire phrase with one of the choices listed.
3. If you see [optional], it means you only add that line if necessary for the task, otherwise remove that line.
4. The environment contains apple, banana, box, bowl, drawer_handle, faucet_handle. Do not invent new objects not listed here.
5. The bowl is large enough to have all other object put in there.
6. I will tell you a behavior/skill/task that I want the manipulator to perform and you will provide the full plan, even if you may only need to change a few lines. Always start the description with [start of plan] and end it with [end of plan].

7. You can assume that the robot is capable of doing anything, even for the most challenging task.
8. Your plan should be as close to the provided template as possible. Do not include additional details.

### v) Reward Coder Prompt for Dexterous Manipulator

We have a plan of a robot arm with palm to manipulate objects and we want you to turn that into the corresponding program with following functions:

```
def set_l2_distance_reward(name_obj_A, name_obj_B)
```

where name_obj_A and name_obj_B are selected from ["palm", "apple", "banana", "box", "bowl", "drawer_handle", "faucet_handle", "drawer_center", "rest_position"]. This term sets a reward for minimizing l2 distance between name_obj_A and name_obj_B so they get closer to each other. rest_position is the default position for the palm when it's holding in the air.

```
def set_obj_orientation_reward(name_obj, x_axis_rotation_radians)
```

this term encourages the orientation of name_obj to be close to the target (specified by x_axis_rotation_radians).

```
def execute_plan(duration=2)
```

This function sends the parameters to the robot and execute the plan for "duration" seconds, default to be 2.

```
def set_joint_fraction_reward(name_joint, fraction)
```

This function sets the joint to a certain value between 0 and 1. 0 means close and 1 means open. name_joint needs to be select from ['drawer', 'faucet']

```
def set_obj_z_position_reward(name_obj, z_height)
```

this term encourages the orientation of name_obj to be close to the height (specified by z_height).

```
def reset_reward()
```

This function resets the reward to default values.
Example plan: To perform this task, the manipulator's palm should move close to object1=apple. object1 should be close to object2=bowl. object2 needs to be rotated by 30 degrees along x axis. object2 needs to be lifted to a height of 1.0.
This is the first plan for a new task.
Example answer code:

```
import numpy as np

reset_reward()
    # This is a new task so reset reward; otherwise we don't need it
set_l2_distance_reward("palm", "apple")
set_l2_distance_reward("apple", "bowl")
set_obj_orientation_reward("bowl", np.deg2rad(30))
set_obj_z_position_reward("bowl", 1.0)

execute_plan(4)
```

Remember:
1. Always format the code in code blocks. In your response execute_plan should be called exactly once at the end.
2. Do not invent new functions or classes. The only allowed functions you can call are the ones listed above. Do not leave unimplemented code blocks in your response.
3. The only allowed library is numpy. Do not import or use any other library.
4. If you are not sure what value to use, just use your best judge. Do not use None for anything.
5. Do not calculate the position or direction of any object (except for the ones provided above). Just use a number directly based on your best guess.
6. You do not need to make the robot do extra things not mentioned in the plan such as stopping the robot.

### vi) Baseline: Code-as-Policies Prompt for Dexterous Manipulator

We have a manipulator and we want you to help plan how it should move to perform tasks using the following APIs:

```
def end_effector_to(position_obj)
```

position_obj is a list of 3 float numbers [x,y,z]

```
def end_effector_open()
```

Open the end effector.

```
def end_effector_close()
```

Close the end effector.

```
def get_object_position(obj_name)
```

Given an object name, return a list of 3 float numbers [x,y,z] for the object position. the object can come from a list of ["apple", "banana", "bowl", "box", "drawer_handle", "faucet_handle", "drawer_center", "rest_position"]

```
def get_normalized_joint_position(joint_name)
```

Given an joint name, return a float numbers x. the joint can come from a list of ["drawer", "faucet"]

```
def reset()
```

Reset the agent.
Example answer code:

```
import numpy as np

reset()
apple_pos = get_object_position("apple")
end_effector_to(apple$_pos)
```

Remember:
1. Always format the code in code blocks.
2. Do not invent new functions or classes. The only allowed functions you can call are the ones listed above. Do not leave unimplemented code blocks in your response.
3. The only allowed library is numpy. Do not import or use any other library.
4. If you are not sure what value to use, just use your best judge. Do not use None for anything.
5. You do not need to make the robot do extra things not mentioned in the plan such as stopping the robot.
6. Try your best to generate code despite the lack of context.

## A.6 Reward functions used in our experiments

In this work we use a set of generic reward functions for each embodiment that the LLMs can modulate. More specifically, we design a set of residual terms as in Equation 1 that are optimized to reach zero by internally converting them to a l2 loss. Thus given a residual term $r(\cdot)$ a reward term can be recovered by $-||r(\cdot)||_2^2$. Below we describe the full set of residual terms we use in our experiments for each embodiment. For each term we select the weights for them to have about the same magnitude. The reward coder can adjust the parameters in each term and optionally set the weight to zero to disable a term.

### A.6.1 Quadruped

Table 3 shows the residual terms used in the quadruped tasks. Note that for the foot-related terms, they are repeated for all four feet respectively. Furthermore, LLMs can optionally set the target foot positions **fp** directly or through a periodic function $max(asin(b2\pi+c),0)$ where $a$ is the magnitude of the motion, $b$ is the frequency, and $c$ is the phase offset.

| Residual Term | Formulation | Default weight |
|---|---|---|
| CoM X-Y position | $\|\mathbf{p}_{xy} - \bar{\mathbf{p}_{xy}}\|$ | 0.3 |
| CoM height | $\mathbf{p}_z - \bar{\mathbf{p}_z}$ | 1.0 |
| base yaw | $\mathbf{p}_{yaw} - \bar{\mathbf{p}_{yaw}}$ | 0.3 |
| base pitch | $\mathbf{p}_{pitch} - \bar{\mathbf{p}_{pitch}}$ | 0.6 |
| base roll | $\mathbf{p}_{roll} - \bar{\mathbf{p}_{roll}}$ | 0.1 |
| forward velocity | $\dot{\mathbf{p}}_x - \bar{\dot{\mathbf{p}}_x}$ | 0.1 |
| sideways velocity | $\dot{\mathbf{p}}_y - \bar{\dot{\mathbf{p}}_y}$ | 0.1 |
| yaw speed | $\dot{\mathbf{p}}_{yaw} - \bar{\dot{\mathbf{p}}_{yaw}}$ | 0.1 |
| foot local position x | $\mathbf{fp}_x - \bar{\mathbf{fp}_x}$ | 1 |
| foot local position y | $\mathbf{fp}_y - \bar{\mathbf{fp}_y}$ | 1 |
| foot local position z | $\mathbf{fp}_z - \bar{\mathbf{fp}_z}$ | 2 |

Table 3: List of residual terms used for the quadruped robot. $\mathbf{p}$ denotes the position and orientation of the robot's torso. $\mathbf{fp}$ denotes the position of the robot's foot (in local space). $\bar{(\cdot)}$ means the target value and $\dot{(\cdot)}$ means the time-derivative of the quantity.

| Residual Term | Formulation | Default weight |
|---|---|---|
| move obj1 close to obj2 | $\|\mathbf{c1}_{xyz} - \mathbf{c2}_{xyz}\|$ | 5 |
| move obj to target X-Y position | $\|\mathbf{c}_z - \bar{\mathbf{c}}_z\|$ | 5 |
| move obj to target height | $\|\mathbf{c}_{xy} - \bar{\mathbf{c}}_{xy}\|$ | 10 |
| move obj to target orientation | $\|\mathbf{o}_{obj} - \bar{\mathbf{o}}\|$ | |
| move joint to target value | $q - \bar{q}$ | 10 |

Table 4: List of reward terms used for the dexterous manipulator robot. $\mathbf{c}$ denotes the position of the object, $\mathbf{o}$ denotes the orientation of the object, $q$ is the degree of freedom to be manipulated.

### A.6.2 Dexterous Manipulator

### A.6.3 Sim-to-Real residual term

As seen in the supplementary video, MuJoCo MPC can discover highly dynamic and dexterous manipulation skills that exceeds the capabilities of existing hardwares. To enable successful deployment on the hardware, we design a regularization term to help achieve steady motions on the robot. Specifically, we use the following residual term:

$$
r_{sim2real} = 3 \begin{cases} \dot{\mathbf{p}}_{ee}, & \text{if } \|\dot{\mathbf{p}}_{ee}\| > 0.3 \\ 0, & \text{otherwise} \end{cases}
$$
$$
+ \begin{cases} \dot{\mathbf{q}}, & \text{if } \|\dot{\mathbf{q}}\| > 0.7 \\ 0, & \text{otherwise} \end{cases}
$$
$$
+ 0.05 \dot{\mathbf{p}}_{obj}
$$
$$
+ 0.1 \begin{cases} \dot{\mathbf{p}}_{ee} - \dot{\mathbf{p}}_{obj}, & \text{if } \|\mathbf{p}_{ee} - \mathbf{p}_{obj}\| < 0.1 \\ 0, & \text{otherwise} \end{cases}
$$
$$
+ 0.4 \begin{cases} \mathbf{q}_{gripper} - 1.0, & \text{if } \|\mathbf{p}_{ee} - \mathbf{p}_{obj}\| > 0.1 \\ \mathbf{q}_{gripper}, & \text{otherwise} \end{cases},
$$

where $\dot{\mathbf{p}}_{ee}$ is the robot end effector velocity, $\dot{\mathbf{q}}$ is the robot joint velocity, $\dot{\mathbf{p}}_{obj}$ is the object velocity. The first two terms regularizes the joint and end effector velocities to encourage them to stay at a low-speed region to prevent jerky robot motions. The third and fourth term encourage objects to move slowly and match gripper speed. This is to discourage dynamic in-hand movements. The last term encourages gripper to open while being away from objects and close when approaching objects. This term helps improve the grasping skill transfer by encouraging a more clean grasp motion.

### A.7 Detailed evaluation results for each task

Figure 10 and Figure 9 shows the full evaluation results for individual tasks in Section 4.4. For each task, we query the *Reward Translator* LLM 10 times. For each generated reward code, we evaluate on MJPC for 50 times and report the success rate. Therefore the results here shows the individual performance of *Reward Translator* and *Motion Controller*. We can see that for most tasks *Motion Controller* achieves

near perfect success rate as long as the right reward is provided, while for tasks that require more dynamic motion or more precise manipulation (e.g. biped standing, and upright banana) *Motion Controller* shows more variance in performance.

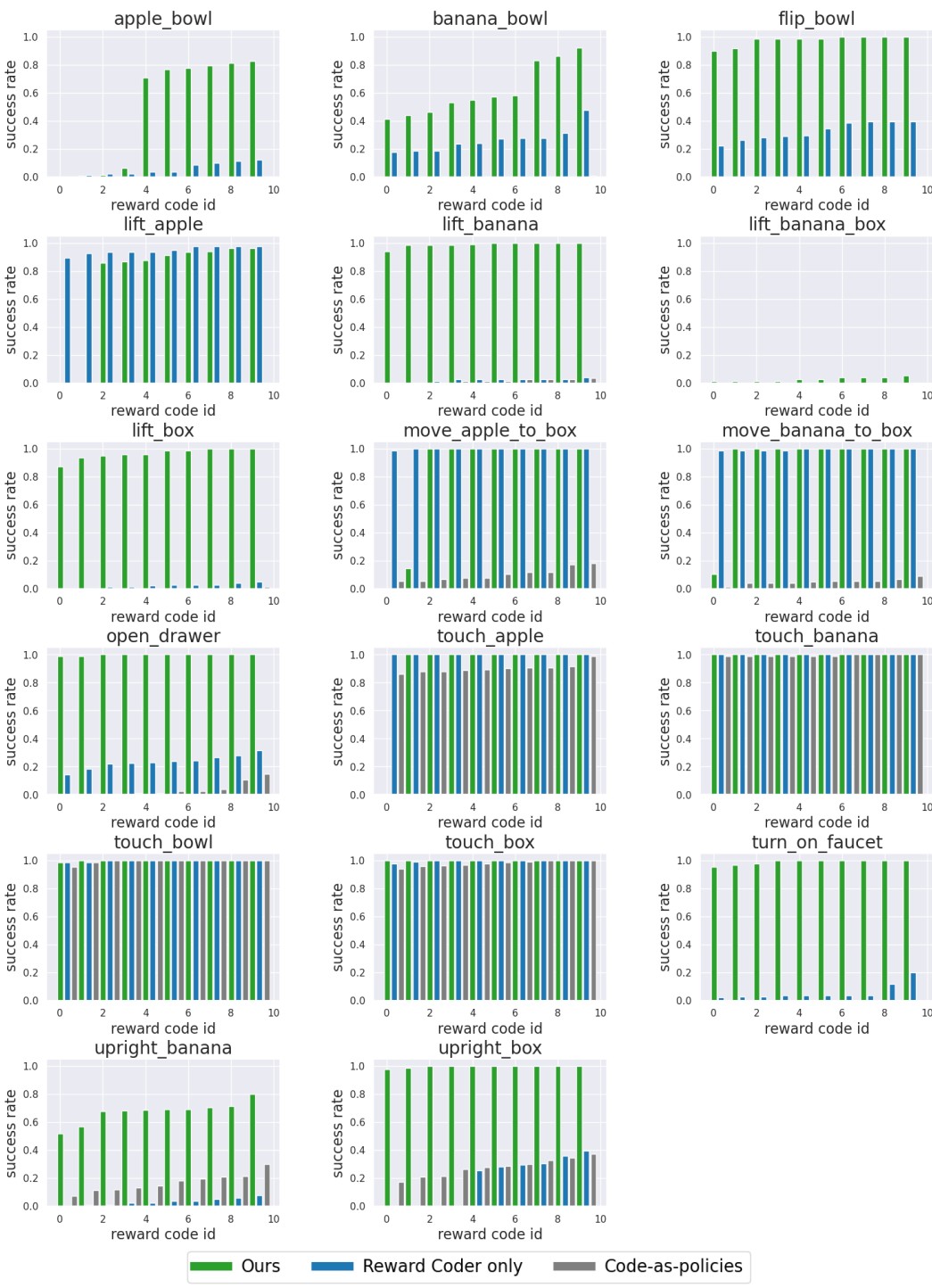

Figure 9: Full evaluation results for the Dexterous Manipulator robot. Note that to better interpret the results, we order the generated reward code in the figures based on the mean success rate.

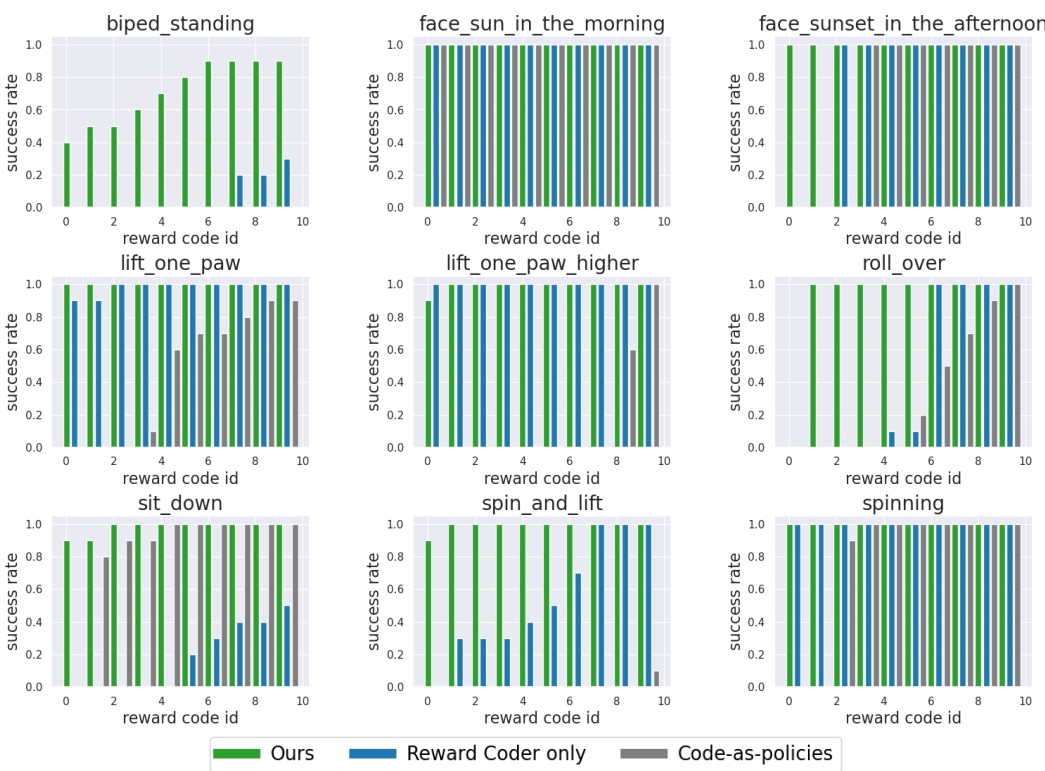

Figure 10: Full evaluation results for the Quadruped robot. Note that to better interpret the results, we order the generated reward code in the figures based on the mean success rate.

## A.8 Example outputs

To help understand the behavior of different methods discussed in this work, here we show sampled responses for two tasks across different methods: *Biped standing* for quadruped, and *Turn on faucet* for dexterous manipulator. For each method, we sample 2-3 responses to show both successful and failed results.

### A.8.1 Sampled results for *Biped standing* task

**User instruction**: *Make the robot stand upright on two back feet like a human.*

**Ours**

```
Sample 1

import numpy as np

reset_reward()
set_torso_targets(0.65,
    np.deg2rad(90), np.deg2rad(0), (0.0, 0.0), None, 0, None)

set_feet_pos_parameters("front_left", 0.65, None, None)
set_feet_pos_parameters("back_left", 0.0, None, None)
set_feet_pos_parameters("front_right", 0.65, None, None)
set_feet_pos_parameters("back_right", 0.0, None, None)

execute_plan()
```

**Sample 2**

```python
import numpy as np

reset_reward()
set_torso_targets(0.65,
    np.deg2rad(90), np.deg2rad(0), (0.0, 0.0), None, 0, None)

set_feet_pos_parameters("front_left", 0.65, None, None)
set_feet_pos_parameters("back_left", 0.0, None, None)
set_feet_pos_parameters("front_right", 0.65, None, None)
set_feet_pos_parameters("back_right", 0.0, None, None)
```

**Sample 3**

```python
import numpy as np

reset_reward()
set_torso_targets(0.65,
    np.deg2rad(90), np.deg2rad(0), (0, 0), None, 0, None)

set_feet_pos_parameters("front_left", 0.65, None, None)
set_feet_pos_parameters("back_left", 0.0, None, None)
set_feet_pos_parameters("front_right", 0.65, None, None)
set_feet_pos_parameters("back_right", 0.0, None, None)

set_feet_stepping_parameters("front_left", 0.0, 0.0, 0.0, 0.0, 0.0, False)
set_feet_stepping_parameters("back_left", 0.0, 0.0, 0.0, 0.0, 0.0, False)
set_feet_stepping_parameters("front_right", 0.0, 0.0, 0.0, 0.0, 0.0, False)
set_feet_stepping_parameters("back_right", 0.0, 0.0, 0.0, 0.0, 0.0, False)

execute_plan()
```

**Reward Coder only**

**Sample 1**

```python
import numpy as np

# Set torso targets to stand on back feet
set_torso_targets(target_torso_height=0.6,
    target_torso_pitch=0.5*np.pi, target_torso_roll=0.0,
    target_torso_location_xy=None, target_torso_velocity_xy=None,
    target_torso_heading=None, target_turning_speed=None)

# Lift front feet off the ground
set_feet_pos_parameters(feet_name="front_left",
    lift_height=0.3, extend_forward=None, move_inward=None)
set_feet_pos_parameters(feet_name="front_right",
    lift_height=0.3, extend_forward=None, move_inward=None)

# Execute the plan
execute_plan(plan_duration=3)
```

```
Sample 2

    import numpy as np

    # Set the torso targets to stand on two back feet
    set_torso_targets(target_torso_height=0.6,
        target_torso_pitch=0.0, target_torso_roll=0.0,
        target_torso_location_xy=None, target_torso_velocity_xy=(0,
        0), target_torso_heading=None, target_turning_speed=None)

    # Lift the front feet off the ground
    set_feet_pos_parameters("front_left",
        lift_height=0.3, extend_forward=None, move_inward=None)
    set_feet_pos_parameters('front_right',
        lift_height=0.3, extend_forward=None, move_inward=None)

    # Execute the plan
    execute_plan(plan_duration=3)
```

**Code-as-Policies**

```
Sample 1

    set_target_joint_angles("front_left", [0, 1, 1.5])
    set_target_joint_angles("back_left", [0, 0.5, 0])
    set_target_joint_angles("front_right", [0, 1, 1.5])
    set_target_joint_angles("back_right", [0, 0.5, 0])
    execute_plan()
```

```
Sample 2

    set_target_joint_angles("front_left", [0, 0.5, 1.5])
    set_target_joint_angles("back_left", [0, 1.2, 0.5])
    set_target_joint_angles("front_right", [0, 0.5, 1.5])
    set_target_joint_angles("back_right", [0, 1.2, 0.5])
    execute_plan()
```

**Sample 3**

```python
import numpy as np

def set_legs_to_pose(pose):
    set_target_joint_angles("front_left", pose[0])
    set_target_joint_angles("back_left", pose[1])
    set_target_joint_angles("front_right", pose[2])
    set_target_joint_angles("back_right", pose[3])

def execute_pose(pose, duration=2):
    set_legs_to_pose(pose)
    execute_plan(duration)

# set initial pose
initial_pose = [
    [0, 0, 0], # front_left
    [0, 0, 0], # back_left
    [0, 0, 0], # front_right
    [0, 0, 0] # back_right
]
set_legs_to_pose(initial_pose)
execute_plan()

# set target pose
target_pose = [
    [0, 0.3, 0.5], # front_left
    [0, 1.2, 1.5], # back_left
    [0, 0, 0], # front_right
    [0, 1.2, 1.5] # back_right
]
execute_pose(target_pose)
```

### A.8.2 Sampled results for *Turn on faucet* task

**User instruction**: *Turn on faucet.*

**Ours**

**Sample 1**

```python
import numpy as np

reset_reward()
    # This is a new task so reset reward; otherwise we don't need it
set_l2_distance_reward("palm", "faucet_handle")
set_obj_orientation_reward("faucet_handle", np.deg2rad(90))
set_joint_fraction_reward("faucet", 1)

execute_plan(2)
```

**Sample 2**

```python
import numpy as np

reset_reward()
    # This is a new task so reset reward; otherwise we don't need it
set_l2_distance_reward("palm", "faucet_handle")
set_joint_fraction_reward("faucet", 1) # 1 means open

execute_plan(2)
```

## Reward Coder only

**Sample 1**

```python
import numpy as np

reset_reward()
set_joint_fraction_reward("faucet", 1)

execute_plan()
```

**Sample 2**

```python
import numpy as np

reset_reward() # Reset reward for a new task
set_joint_fraction_reward("faucet",
    1) # Turn on the faucet by setting the joint fraction to 1 (open)

execute_plan(2)
```

## Code-as-Policies

**Sample 1**

```python
reset()
faucet_handle_position = get_object_position("faucet_handle")
end_effector_to(faucet_handle_position)
end_effector_close()
normalized_joint_position = get_normalized_joint_position("faucet")
end_effector_open()
```

```python
#python
import numpy as np

reset()
faucet_handle_pos = get_object_position("faucet_handle")
end_effector_to(faucet_handle_pos)
end_effector_open()
end_effector_close()

normalized_faucet_position = get_normalized_joint_position("faucet")
target_faucet_position = np.clip(normalized_faucet_position + 0.5, 0, 1)

end_effector_to([faucet_handle_pos[0],
    faucet_handle_pos[1], faucet_handle_pos[2] + 0.1])
end_effector_to([faucet_handle_pos[0],
    faucet_handle_pos[1], faucet_handle_pos[2]])
end_effector_open()
```

## A.9 Full outputs for the interactive examples

Below we show the output reward code from our system for the two interactive examples: tasking a quadruped robot to perform moonwalk, and tasking a dexterous manipulator robot to place an apple in a drawer.

**Quadruped moonwalk**

Instruction 1: Make the robot stand upright on two back feet like a human.

```python
import numpy as np

reset_reward()
set_torso_targets(0.65,
    np.deg2rad(90), np.deg2rad(0), (0.0, 0.0), None, 0, None)

set_feet_pos_parameters("front_left", 0.65, None, None)
set_feet_pos_parameters("back_left", 0.0, None, None)
set_feet_pos_parameters("front_right", 0.65, None, None)
set_feet_pos_parameters("back_right", 0.0, None, None)

execute_plan()
```

**Instruction 2:** Good, you actually don't need to keep the front paws at certain height, just leave it to the controller.

```python
import numpy as np

reset_reward()
set_torso_targets(0.65,
    np.deg2rad(90), np.deg2rad(0), (0.0, 0.0), None, 0, None)

set_feet_pos_parameters("back_left", 0.0, None, None)
set_feet_pos_parameters("back_right", 0.0, None, None)

# Set stepping parameters for all feet to disable stepping motion
for foot in ["front_left", "rear_left", "front_right", "rear_right"]:
    set_feet_stepping_parameters(foot, 0.0, 0.0, 0.0, 0.0, 0.0, False)

execute_plan()
```

**Instruction 3:** Good, now make the robot do a moon-walk while stand up like this.

```python
import numpy as np

reset_reward()
set_torso_targets(0.65, np.deg2rad(90), np.deg2rad(0), None, ( -0.2 , 0.0),
    0, None)

set_feet_pos_parameters("back_left", None, None, None)
set_feet_pos_parameters("back_right", None, None, None)

set_feet_stepping_parameters("back_left", 0.5, 0.5, 0.5, 0.1, -0.2, True)
set_feet_stepping_parameters("back_right", 0.5, 0.5, 0.0, 0.1, -0.2, True)
set_feet_stepping_parameters("front_left", 0.0, 0.0, 0.0, 0.0, 0.0, False)
set_feet_stepping_parameters("front_right", 0.0, 0.0, 0.0, 0.0, 0.0, False)

execute_plan()
```

**Instruction 4:** Now close the drawer.

```python
import numpy as np

reset_reward()
    # This is a new task so reset reward; otherwise we don't need it
set_l2_distance_reward("palm", "drawer_handle")
set_joint_fraction_reward("drawer", 0)

execute_plan()
```

