# OpenReview forum: "Language to Rewards for Robotic Skill Synthesis"
_robot-learning.org/CoRL/2023/Conference — CoRL 2023 Oral_

### Official Review · Reviewer_NCUb · 2023-07-17

**Confidence:** 4
**Originality:** Good
**Technical Quality:** Very Good
**Clarity Of Presentation:** Very Good
**Impact:** 3

**Recommendation:**

Weak Accept: I recommend accepting the paper, but will not argue for my recommendation if the majority of other reviewers have a different opinion.

**Review:**

Large language models have demonstrated remarkable capabilities in natural language understanding. Nevertheless the direct application of LLMs to control robotic devices is not possible. The authors investigate one of the possible approaches that generates reward functions from LLM given a textual description of a task. This approach seems to be quite generic since in a wide range of applications reward can be defined by simple equation. The paper clearly presents the developed approach and the evaluation results. The authors show that the proposed method performs better then code-as-policies baseline and baseline with only one LLM with returns reward directly from natural language task description.

One possible limitation of the presented approach is simplicity of generated reward function. For example to train a robust reinforcement learning policy to control quadruped robot on needs to design a reward function with multiple terms punishing big torques, velocities, non-smooth movements. However the punishment terms may be similar for different tasks and may be handcrafted and shared between them.

Also there are some sentences with missing spaces like "... the Reward Translatorthat interprets the user ..."

**Quality Of The Limitations Section:**

Limitations are addressed clearly

**Questions For Rebuttal:**

How easy it would be to apply this approach to generate different motion styles like gallop, trot, walk?

**Robotics Focus:**

Sufficient demonstration on hardware

**Summary Of Paper:**

In the present work, the authors explore a task to employ LLM in robotics. In their approach, LLM defines a task by generating a reward function which is afterward optimized by model predictive control methods. The reward function is generated in two steps. In the first step, LLM receives a description of the desired state and outputs the target position of the robot. In the second step, LLM receives a description of the target position and outputs the reward function. The authors demonstrate the capabilities of the proposed method in a simulated quadruped robot and a dexterous manipulator robot and on a real robotic arm.

**Summary Of Recommendation:**

I recommend accepting the paper. It presents an interesting method to ground LLM for robotics tasks.

---

### Official Review · Reviewer_ba4p · 2023-07-20

**Confidence:** 3
**Originality:** Good
**Technical Quality:** Good
**Clarity Of Presentation:** Excellent
**Impact:** 3

**Recommendation:**

Weak Reject: I recommend rejecting the paper, but will not argue for my recommendation if the majority of other reviewers have a different opinion.

**Review:**

**Significance:**
The authors propose a two-part prompt for GPT-4 to produce parameterized reward code. While this approach is interesting, I believe it lacks novelty. Much of the contribution seems to center around the prompt engineering to produce a reward parameterization. Afterwards, the paper uses a standard motion predictive controller. However, I believe that the paper does explore an important avenue within LLM + Robotic Learning / AutoRL research area.

I believe the paper would significantly benefit from evaluating the method’s performance on a vaguer “motion descriptor prompt” and a “reward coder prompt” standardized across robot types. The current method, as-is, still requires extensive domain expertise and human intervention to generate reward code.

**Quality:**
The experiments conducted by the authors were thorough and well-designed within the paper’s objective. The authors clearly explored two RL domains (quadruped robot, serial manipulator) through 17 total tasks. As an additional ablation study, I would be curious to see the RL performance when the LLM is given varying levels of context (ex: fewer variable definitions, fewer reminders, etc).

I would also be interested in seeing results from the real-world experiment. Were the success rates for each task similar to the simulated rates?

**Clarity:**
The paper was very well written and clear. The data was presented and explained in a direct and concise manner. The sample prompts and videos in the supplemental materials were well received. There were a few spelling and grammar errors that do not impact the overall message of the paper.
- Figure 1 caption requires space between "Translator" and "that".


As summarized from the above discussion on significance, quality, and clarity...

**Strengths:**
- The paper was clear and concise. The supporting materials were helpful.
- The experimental design allowed for a thoroughly exploration of the scope of the paper.
- The exploration of interactive systems for reward tuning was a cool idea.

**Weaknesses:**
- No sensitivity study of the effects of varying the amount of context given to the LLM.
- Although interesting, the methods in the paper only produce a incremental contribution.
- Experiments largely focus on simulated results. Quantitative results on real world experiments should be included.


**Quality Of The Limitations Section:**

Limitations are addressed clearly

**Questions For Rebuttal:**

Please address the concerns in the review. No additional questions.

**Robotics Focus:**

Sufficient demonstration on hardware

**Summary Of Paper:**

The paper introduces a pipeline for using LLM to generate reward specification for a variety of RL tasks for a quadruped and serial robot manipulator. The authors design a two-part pipeline to prompt GPT-4 to create reliable and parameterized reward code used to train the motion controller using a MJPC planner. The work conducts an ablation study using different prompting contexts to demonstrate differences in the RL model’s success rate on each task.

**Summary Of Recommendation:**

While the results are interesting and the experiment selection was well designed, I believe the paper still lacks in several key areas. I do not feel that the paper is significantly novel, and there is a lack of real world quantitative results. As a result, I would recommend a weak rejection for this paper.

---

### Official Review · Reviewer_8cwu · 2023-07-21

**Confidence:** 4
**Originality:** Fair
**Technical Quality:** Good
**Clarity Of Presentation:** Good
**Impact:** 3

**Recommendation:**

Weak Accept: I recommend accepting the paper, but will not argue for my recommendation if the majority of other reviewers have a different opinion.

**Review:**

The authors hypothesize that producing rewards, instead of directly producing robot commands, is more natural for LLMs and is likely to make better use of their inductive bias. Their experiments on simulated quadrupeds, an arm with dexterous hand, and a real-world robot equipped with parallels jaw gripper seem to back this up. Results seem promising compared to a code-as-policy baseline and their implementation detail of making reward prediction a two step process also appears affective when compared to the baseline of Reward Coder only.

That being said, I have some questions, likely beyond the scope of this paper, about this approach scaling. The tasks used here were all fairly simple, which means they were amenable to solving via MPC. That's not to say that LLM reward generation wasn't useful, it certainly eased the design work of humans requesting tasks, but for more complex tasks finding good rewards that both sufficiently constrain behavior and allow efficient solutions is known to be difficult and this work does not address that difficulty.

The work is also lacking on details about the LLM used. It seems it was an off-the-shelf LLM with only in-context learning used.

As an aside, this seems highly related to the concurrent work VoxPoser (https://voxposer.github.io/). That's not itself a problem, but the authors should consider citing it as concurrent work.

**Quality Of The Limitations Section:**

Limitations are addressed clearly

**Questions For Rebuttal:**

- What LLM was used for the Motion Descriptor and Reward coder and was any finetuning performed?


**Robotics Focus:**

Sufficient demonstration on hardware

**Summary Of Paper:**

The authors propose using a two-step Code-As-Policy approach using LLMs. This two step approach first translates a natural language prompt into a set of rewards and then translates those rewards into a set of function calls to be executed. Under the hood, mujoco MPC is used to turn the rewards into a plan which is executed on the robot.

**Summary Of Recommendation:**

While somewhat limited, the work illustrates an interesting and potentially useful factorized prompting strategy for generating rewards from an LLM that are of sufficient quality to synthesize (simple) behaviors. That's a useful, if slightly incremental, finding.

---

### Author Response · Authors · 2023-08-09
**Response to all reviewers**

We would like to thank all reviewers for the time and effort to provide insightful and constructive feedback for our work.

We have revised the manuscript (shown in blue text in the updated pdf file) according to reviewer feedback and replied to each reviewer. Below we provide a summary of the revision we have made.

* Included a new set of tasks for generating different types of gaits of the quadruped robot (without modifying any of the prompts). We demonstrated that our system can be directly applied to generate three types of gaits (trotting, pacing, bounding) with high success rate. We have included videos of the simulated robot in the attached file for each reply.
* Included an ablation study on the amount of context/reminders provided to both Motion Descriptor and Reward Coder to analyze the sensitivity of the system. Our ablation study shows improved performance with our full prompt setup, which can be found in Section 4.5 of the revised manuscript. (please find revised manuscript in the attachment for each reply)
* Incorporated latest work in this direction in the related work section.
* Added missing details, fixed typos.

We hope these changes can help mitigate the concerns/questions by the reviewers and please let us know if there are additional questions!

---

> ### Author Response · Authors · 2023-08-12
> **Updated real robot experiments**
>
> Dear reviewers,
>
> To further support the real-world performance of our proposed approach, we have conducted additional real experiments and collected quantitative results. We have also improved the deployment pipeline that significantly increased the robot’s efficiency. We evaluated our pipeline on three tasks: picking up a cube, picking up an apple, and opening a drawer. For each task, we run our pipeline 10 times and report the success rate as shown below. We also want to note that the failed trials were due to sim-to-real gaps in the system such as noises in the identified object pose, which go beyond the scope of language to reward study in this paper, and are in our plan for further investigation.
> | Pick up cube | Pick up apple | Open the middle drawer |
> |-----------|-----------|-----------|
> | 0.7 | 0.7 | 0.8|
>
> We have updated our project website https://language-to-reward-review.github.io/ with the latest real-world experiment videos. For easier comparison here are the links to the [previous video](https://language-to-reward-review.github.io/videos/real/l2r_real.mp4) and the [updated video](https://language-to-reward-review.github.io/videos/real/l2r_demo.mp4) with improved deployment pipeline.
>
> We hope this demonstrates the applicability of our approach to real robots and address potential concerns. Meanwhile, please let us know if you have additional questions/comments about our work!

---

> > ### Author Response · Authors · 2023-08-14
> > **Additional feedback/questions are welcome!**
> >
> > Dear reviewers,
> >
> > Thanks again for your insightful reviews! Please let us know if you have additional questions or concerns regarding our work and we'd be happy to provide more discussion and details!
> >
> > - Authors

---

### Decision · Program_Chairs · 2023-08-30

**Decision:**

Accept (Oral)

**Comment:**

This paper addresses the problem of skill generation where an input language instruction must be translated to a reward model. The core contribution lies in the design of a reward interface that associates a high level instruction to parameters of a reward model. Once the reward model is known, a MPC is used to generate control. From the LLM perspective, the approach lowers the symbolic to real boundary by reasoning over reward parameters instead of pure symbols as used in task planning.

During the rebuttal phase the authors provided clarifications on the experiments, engaged with closely related works and uploaded data on real world experiments. The authors are requested to update their manuscript based on the reviewer feedback.